# Axiomatic Explainer Locality with Optimal Transport

## Abstract

Explainability methods have been notoriously difficult to evaluate and compare. Because of this, practitioners are often left guessing as to which explainer they should use for their task. *Locality* is one critical property of explainers which grants insight into the diversity of produced explanations. In this paper, we define a set of axioms which align with natural intuition regarding *globalness*, the inverse of locality. We then introduce a novel measure of globalness, Wasserstein Globalness, which uses optimal transport to quantify how local or global a given explainer is. Finally, we provide theoretical results describing the sample complexity of Wasserstein Globalness, and experimentally demonstrate how globalness can be used to effectively compare explainers. These results illustrate connections between both explainer fidelity and explainer robustness.

## 1 Introduction

Machine Learning (ML) models are increasingly complex and capable of impressive performance in several domains. However, as models become more complex, they also become less interpretable. For this reason, researchers have begun to explore the topic of explainability, where model decisions are assigned an *explanation*. These explanations come in many forms, but often indicate how important each feature is towards the model prediction.

Explainers need to be trustworthy for their explanations to be valuable. However, ML practitioners have very little information at their disposal when deciding which explainer is right for them. Unlike traditional ML models, whose accuracy can be computed on held-out testing datasets, there is no obvious metric by which we can compare explainers. Ground-truth explanations are rarely known, meaning we cannot directly compute the accuracy of an explainer.

Some authors have tried to argue that their explainer is best by proposing a variety of pseudo-accuracy metrics. When explaining image classifiers, for example, one may follow the lead of Zhang et al. (2018) and Wang et al. (2020), by evaluating explainers based on how well they concentrate their saliency map's energy around the object of interest. This kind of evaluation-metric is far from perfect, as it penalizes explainers from using scene context and is heavily biased towards concentrated saliency. In addition, these psuedo-accuracy metrics are only applicable for the specific task of feature-attribution for image data, when in reality, there are many other types of explanations and many other types of data. Clearly, it would be of great interest to the explainability community to be able to compare and contrast general-purpose explainers for any ML task.

*Globalness* is a property that can be used to compare and contrast explainers. When one explanation fully explains the model's behavior, we call this a *global explanation*. In the past, models were typically explained globally. For example, feature selection would be done globally, meaning it would generate a single group of salient features for the entire dataset (Song et al., 2010; Yu & Liu, 2004; John et al., 1994; Dy & Brodley, 2004). More recently, in order to explain complex black-box models, it has become common to generate instance-wise explanations rather than a single explanation for the entire model. In this case, the explainer outputs *local explanations* which apply to only a subset of the model inputs. Since this distinction between local explainers and global explainers emerged, researchers have begun to acknowledge locality/globalness as a property of explainers.

Globalness is a meaningful property of explainers because it indicates how uniform the explanations are. In some cases, we expect or even desire all explanations to be similar to one another. Globalness

is also related to the concept of stability and robustness. The robustness of an explainer is limited by its globalness, and we can even use an explainer's globalness to measure its local-robustness, since an explainer that is robust to small changes in the input will be near-global in a local neighborhood of the input space. This is another way in which a measure of globalness can facilitate a better understanding of our explainers, and allow us to compare/contrast their behavior.

In this paper, we study the property of globalness. We provide a novel way to measure it, and present several examples of how to use it in practice. To our knowledge, this is the first work that has provided a formal measure for explainer locality/globalness. We believe that this advances the field by enabling a more thorough analysis of various explanation techniques. With our measure of globalness, one can compare and contrast explainers in a way that was previously difficult and heuristic.

The contributions of the paper are as follows. First, we introduce intuitive axiomatic properties which align with human intuition surrounding the notion of locality/globalness. Second, we propose *Wasserstein Globalness*, a novel measure of globalness, that satisfies all these properties. We also present theoretical results regarding the sample complexity of estimating Wasserstein Globalness. Finally, through our experiments, we demonstrate how explainers can be differentiated by their globalness, and make a connection between globalness and adversarial robustness.

## 2 RELATED WORK

As mentioned before, the community has struggled to effectively evaluate explainers, since real-world data rarely comes with ground-truth explanations. For this reason, several authors have turned to synthetic data, where the relevance of features are known (Chen et al., 2018; 2017). Instead of accuracy, others seek to describe explainer properties like transparency, sparsity, or robustness, in order to inform ML practitioners' choice of explainer (Zhang et al., 2021). Because of the interaction between the data and explainer, we often want to describe properties of an explainer when applied to a specific data sample, like locality. A measure of globalness which accounts for both the data and explainer would provide a quantitative method for comparing and contrasting different explainers.

Several authors have discussed the locality/globalness of explainers. For example, Doshi-Velez & Kim (2017) describe two types of explainers: local and global. They claim that global explainers are useful for scientific understanding or bias detection, and local explainers are useful for understanding specific model predictions. Zhang et al. (2021) surveys the interpretability literature and categorizes explainers along several axes. One of these axes is related to the locality of the explainer, where explainers are either "global", "local", or "semi-local". The distinctions provided by Doshi-Velez & Kim (2017); Zhang et al. (2021) are categorical rather than continuous. Though the community has begun to consider this property, there is still no formal continuous measure of locality/globalness.

One common explainer, LIME, requires the user to specify a kernel width which is roughly related to locality/globalness (Ribeiro et al., 2016). Anchors are a rule-based explanation given by a constrained optimization problem, where the objective is the "coverage" of the anchor (Ribeiro et al., 2018). An anchor's coverage indicates how broadly applicable the rule is, thus this is also directly related to locality/globalness, but is tied directly into the objective of the explainer.

Some authors, like Ribeiro et al. (2016), offer ways to construct a single global explanation from many local explanations, thereby granting a higher-level understanding of the model. While this offers explanations at two levels of globalness, we emphasize that this is again only a binary distinction. While works like this acknowledge the property of locality and its importance to explainability, they offer no way to quantify the locality of an explainer.

The need for a measure like this is becoming increasingly apparent as more researchers study the theory of explainability. For example, Li et al. (2020) define an object called the neighborhood disjointedness factor in order to study the generalization of finite-sample based local approximation explainers. Neighborhood disjointedness measures roughly how far apart points are from one another, and could be applied to explanations as a measure of globalness. However, this is limited to a small class of explainers, and does not apply to general explanation frameworks.

We advance the existing literature by formalizing the property of explainer globalness, and proposing a method for measuring it in practice. This is a new tool that the machine learning community can

use to better understand how our explainers interact with our data. We can order explainers by their globalness, identify grouped explanations, and compare explainers.

## 3 METHOD FORMULATION

There are a number of different ways one could measure the locality/globalness of an explainer. In order to guarantee the quality of our approach, we will first discuss the properties which should be satisfied by *any* measure of explainer globalness. In Section 3.2, these will be stated as axioms which align with our intuition about globalness. In Section 3.3, we will briefly review why other reasonable approaches fall short according to these properties. Then, in Section 3.4, we will give our definition of globalness and demonstrate that it satisfies all of these axioms.

### 3.1 NOTATIONS

Consider a black-box prediction model $f : \mathcal{X} \to Y \subseteq \mathbb{R}^c$, where $\mathcal{X} \subseteq \mathbb{R}^d$ is the space of model inputs, $Y$ is the target, $d$ is the dimension of the data, and $c$ is the number of class labels. An *explainer* for this model is a function $E : \mathcal{X} \to \mathcal{E}$, where the outputs of the explainer are called *explanations*, and comprise the space $\mathcal{E}$. The space $\mathcal{E}$ is either a subset of $\mathbb{R}^d$ for feature-attribution explainers, or the vertices of a hypercube, $\{0,1\}^d$ for feature-selection explainers. We will use $P(\mathcal{E})$ to denote the set of probability distributions over the space of explanations, $\mathcal{E}$. Let $\mu \in P(\mathcal{E})$ be one such probability distribution, generated by passing the data through our explainer. A metric measure space (mm-space) is a 3-tuple $(\mathcal{E}, d_{\mathcal{E}}, \mu)$, where $\mathcal{E}$ is the space of explanations, and $d_{\mathcal{E}}$ is a distance metric between explanations.

| Explanation Framework | $\mathcal{E}$ | Distance Measure |
|:---:|:---:|:---:|
| Feature Attribution | $\mathbb{R}^d$ | Cosine |
| Feature Selection | $\{0,1\}^d$ | Hamming |

Table 1: Recommended metric spaces for common explanation frameworks.

For $p \in P(X)$ and $q \in P(Y)$, $\Pi(p, q)$ will denote the set of joint distributions in $P(X \times Y)$ which marginalize to $q$ and $p$. $\delta_x$ will denote the Dirac measure centered at $x$, which takes the value 1 at $x$ and 0 elsewhere. We will denote the *push-forward measure* of $\mu$ by $\phi$ as $\phi_{\#}\mu(A) = \mu(\phi^{-1}(A))$. For ease of reference, a summary of the notation used in this paper is in Appendix A Table 2.

### 3.2 DESIRED PROPERTIES OF A GLOBALNESS MEASURE

In this section, we introduce the desired properties of a general globalness measure $G : \mathcal{P}(\mathcal{E}) \to \mathbb{R}$, which assigns a real value to a distribution of explanations. Our properties state that measure should be non-negative, continuous, convex, and should encode similarity between explanations.

**Property 1** (Non-negativity) For all mm-spaces $(\mathcal{E}, d_{\mathcal{E}}, \mu)$, the globalness is non-negative, i.e., $G(\mu) \geq 0$.

**Property 2** (Fully-local measure) Let $U$ be the uniform distribution on $\mathcal{E}$. Then

$$\mu = U \iff G(\mu) = 0 \tag{1}$$

Property 2 states that the uniform distribution gives zero globalness, and Property 1 states that this is the minimum-globalness distribution for a given $(\mathcal{E}, d_{\mathcal{E}})$.

**Property 3** (Fully-global measure) A fully global explainer is one which produces the same explanation for every input, meaning its distribution of explanations can be represented by a Dirac distribution $\delta_{x_0}$. Formally, there exists an $x_0 \in \mathcal{E}$ for which $G_p(\delta_{x_0}) \geq G_p(\mu)$ for all $\mu \in \mathcal{P}(\mathcal{E})$.

**Property 4** (Continuity) Let $\{\mu_n\}$ be a sequence of probability distributions which converge in distribution to $\mu$. Then $G_p(\mu_n) \to G_p(\mu)$.

This property claims that the globalness should be continuous with respect to the input distribution. In practice, the underlying distribution $\mu$ is often unknown, and we need to approximate it with an empirical distribution $\mu_n$, where $n$ is the number of samples. Property 4 guarantees that $G$ can

be approximated by sampling from $\mu$ and finding the globalness of an empirical distribution $\mu_n$. Theorem 2, discussed in Section 4, will characterize the rate of this convergence in more detail.

**The need to account for distance between explanations.** Before we introduce the next property, we will motivate it by discussing the geometry of $\mathcal{E}$. A measure of globalness should take into account a notion of difference between explanations, this can be captured by introducing a distance metric, $d_{\mathcal{E}}$. For an illustrative example as to how globalness is related to the geometry of explanations, refer to Figure 1. Both distributions yield one of two explanations with equal probability. These two explanations are at a hamming distance of 1 for the left explainer, and a distance of two for the right explainer. Since the explanations are more varied for the right explainer, it is less global.

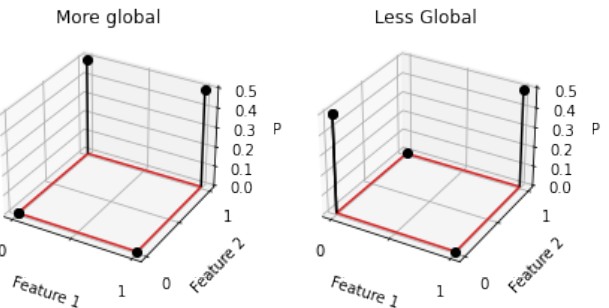

Figure 1: Two distributions of feature-selection explanations on $\{0, 1\}^2$ (red), the vertical axis represents the probability of seeing that explanation. The left distribution is more global because it outputs explanations which are closer to one another in the metric space $(\mathcal{E}, d_{\mathcal{E}})$.

**The need for isometry invariance.** The previous example demonstrated that globalness should account for the distance between explanations. Additionally, when these explanations change, but the distances between them do not, the globalness should not change. In Figure 2, we show an example of this. The two distributions in this figure are equivalent up to a re-ordering of the features. Clearly, we would consider both of these to be equally global, because one distribution is simply a reflection of the other. Transformations like this, rotations and reflections, are referred to as isometries. The following property states that a measure of globalness should be invariant to isometries.

**Property 5** (Isometry-invariance) Let $T_{\mathcal{E}, d_{\mathcal{E}}}$ be the group of distance-preserving measurable maps $\phi$. Then,

$$T_{\mathcal{E}, d} = \{\phi : \mathcal{E} \to \mathcal{E} | d_{\mathcal{E}}(x, y) = d_{\mathcal{E}}(\phi(x), \phi(y)), \forall x, y \in \mathcal{E}\} \quad (2)$$

Then $G$ is invariant with respect to the group $T_{\mathcal{E}, d_{\mathcal{E}}}$:

$$G(\mu) = G(\phi_\# \mu) \text{ for all } \phi \in T_{\mathcal{E}, d_{\mathcal{E}}} \quad (3)$$

Additionally, let $S$ denote all the measurable bijective maps $\psi : \mathcal{E} \to \mathcal{E}$. Then $G$ is *not* invariant to $S$:

$$\text{There exists some } \psi \in S \text{ for which } G(\mu) \neq G(\psi_\# \mu) \quad (4)$$

This axiom states that globalness is invariant to isometries of the explanations $\mathcal{E}$, but not arbitrary permutations. First, Equation 3 stated that $G$ should be invariant to distance-preserving transformations of the explanation-space. Because globalness measures roughly how similar the set of explanations are to one another, any rigid transformation (rotation, reflection, etc.) of the space of explanations will have no effect on the globalness. Recall that we want globalness to measure roughly how similar the explanations are. This notion of similarity is directly encoded by the distance $d_{\mathcal{E}}$, hence an isometry which does not change these distances should not change the globalness.

There is one more situation that we want our properties to address: *when two sets of explanations are combined, how does the globalness change?* The following property answers this question.

**Property 6** (Convexity) If the explanations from two explainers are combined, the combined distribution will yield a lower globalness than the average globalness of the original explainers. Formally, let $R = \lambda P + (1 - \lambda)Q$ for some $\lambda \in [0, 1]$ and $P, Q \in P(\mathcal{E})$. Then $G(R) \leq \lambda G(P) + (1 - \lambda)G(Q)$. This property states that $G$ is convex with respect to distribution $\mu$, implying that a mixture of two explainers is less global than the average of each explainer's globalness. By mixture, we mean probabilistically generating explanation $E_1(x)$ with probability $p$ or generating explanation $E_2(x)$

Figure 2: An example of two sets of explanations. Though the distributions are different, they can be obtained from one another by simply swapping the ordering of the features. This corresponds to a reflection, which is an isometric transformation.

with probability $(1 - p)$ independently for each $x \in \mathcal{X}$. Concavity is a common requirement of diversity measures. For example in genetics, where biological diversity is measured, the diversity of two populations combined is expected to be greater than the diversity among one population (Ricotta, 2003). We instead require convexity because globalness is inversely related to the diversity of explanations.

### 3.3 CANDIDATE MEASURES

We have considered several "candidate" approaches for $G$. We will now discuss some of these, and illustrate why they fall short as a measure of globalness.

**Shannon Entropy.**   Since globalness is related to how diverse the explanations are, one might initially consider using the Shannon Entropy of the explainer's outputs as a measure of locality (Shannon, 1948).

$$H(X) = \mathbb{E}\big[ -\log(p(X)) \big] \tag{5}$$

Entropy is maximized when all the outcomes are equally likely, and minimized when the explainer only generates one explanation for all of the data, clearly satisfiying Properties $1 - 3$. In fact, entropy satisfies several of our desiderata in Section 3.2, but fails on Property 5. Entropy cannot distinguish between how similar explanations are to one another.

**F-Divergences.**   One might also consider measuring globalness with an f-divergence (Csiszár, 1964) between the distribution of explanations $\mu$, and the uniform distribution, $U$:

$$D_f(P\|Q) = \mathbb{E}_{x \sim Q}\big[f(\frac{P(x)}{Q(x)})\big] \tag{6}$$

**Remark** (Alternative approaches). *Shannon entropy and f-divergences both violate Property 5.*

F-divergences fail for the same reason that entropy failed. They consider only the probability of each explanation and ignore the degree of similarity between them. The proof of this remark is given in Appendix E.

### 3.4 THE PROPOSED APPROACH

The framework presented in this paper is very general. Only a metric space $(\mathcal{E}, d_{\mathcal{E}})$ needs to be defined. For common explanation frameworks, recommended choices for $d_{\mathcal{E}}$ are listed in Table 1. Given a metric space $(\mathcal{E}, d_{\mathcal{E}})$, we propose to measure the p-Wasserstein Globalness of $\mu$, denoted $G_p(\mu)$, as the Wasserstein distance between $\mu$ and the uniform measure on $\mathcal{E}$ (Kantorovich, 1960). Intuitively, the Wasserstein distance metric between distributions $P$ and $Q$ measures how much "work" one would have to do to move probability mass from $P$ until it matches $Q$. For this reason, it is sometimes called the *earth-movers distance* when $p = 1$.

**Definition 1** (Wasserstein Globalness).

$$G_p(\mu) := d_W^p(\mu, U) \tag{7}$$

$$= \inf_{\pi \in \Pi(\mu, U)} \mathbb{E}_{(x,y) \sim \pi}[d_{\mathcal{E}}^p(x,y)]^{1/p} \tag{8}$$

For all of our experiments, we will have $p = 1$, but $p$ is left general in the remainder of the paper, with the exception of Theorem 2, where $p$ affects the convergence of approximate solutions to Equation 7.

**Theorem 1.** *The Wasserstein Globalness $G_p : \mathcal{P}(\mathcal{E}) \to \mathbb{R}$ satisfies Properties 1-7. Where $\mathcal{P}(\mathcal{E})$ is the set of probability distributions over $\mathcal{E}$.*

The proof of Theorem 1 is provided in Appendix C. Because it satisfies these intuitive properties, the Wasserstein Globalness is a satisfactory way to quantify the globalness of an explainer.

## 4 SAMPLE COMPLEXITY

In practice, we do not work with the true distributions $\mu$ and $U$, but their discrete approximations. The discrete approximation for a distribution $p$ is defined as $p_N = \frac{1}{N} \sum_{i=1}^N \delta_{X_i}$ for $X_i \overset{\text{iid}}{\sim} p$.

Using a discrete approximation for the uniform distribution allows us to estimate the Wasserstein Globalness with a discretized formulation:

**Definition 2** (Empirical Wasserstein Globalness).

$$\hat{G}_p(\mu) := d_W^p(\mu, U_N) \tag{9}$$

In this section we address the question of how accurately $\hat{G}_p(\mu_N)$ approximates $G_p(\mu)$. Fortunately, there are related works that have studied the convergence of empirical distributions under the Wasserstein metric, notably by Fournier & Guillin (2015); Dereich et al. (2013). These prior results provide the foundation for the following theorem.

**Theorem 2.** *Let $p$ be the order of the Wasserstein Globalness, and let $M_q(\mu) = \int_{\mathbb{R}^d} |x|^q d\mu(x)$. If $p \in (0, d/2)$, $q \neq d/(d-p)$, $q \geq \frac{dp}{d-p}$, then:*

$$\mathbb{E}[|\hat{G}_p(\mu_N) - G_p(\mu)|] \leq \kappa_{p,q} M_q(\mu)^{1/q} N^{-1/d} + C_{p,q,d} M_q^{p/q}(\mu)(N^{-p/d} + N^{-(q-p)/q}) \tag{10}$$

From this theorem, we can immediately see that $\mathbb{E}\big[|G_p(\mu) - G_p(\mu_N)|\big] \to 0$ as $N \to \infty$. However, increasing $d$ will lead to a looser bound, meaning we need more samples as N increases. This is a problem if we want to compute Wasserstein Globalness in high-dimensional problem settings like computer vision tasks. To scale to high dimensions, we utilize the Sinkhorn Distance (Cuturi, 2013), which is described in Appendix B.

## 5 EXPERIMENTS

Wasserstein Globalness allows us to compare and contrast explainers. In this section, we demonstrate certain insights it can provide about both our data and our explainer. In this section, we will present two kinds of experiment. The first demonstrates how globalness can help identify clustered explanations. The second type of experiment uses globalness to identify how locally-stable an explainer is. The code used in all following experiments will be made publicly available.

**Datasets.** Our first experiment uses a simple synthetic dataset where the ground-truth explanations are known. To create this dataset, we generate $d$ Gaussian clusters in $\mathbb{R}^d$. Each cluster has a single (known) relevant feature. We use two other datasets in these experiments, which contain high-dimensional image data. The first, MNIST (Deng, 2012), is comprised of grayscale images depicting handwritten digits 0-9. The last dataset, CIFAR-10 Krizhevsky (2009), is comprised of 32x32 color images. The targets in CIFAR-10 are one of ten classes of common objects.

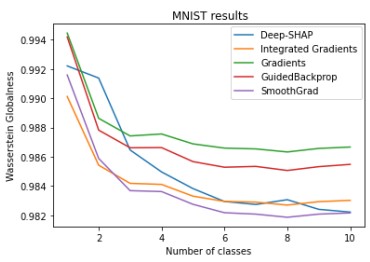 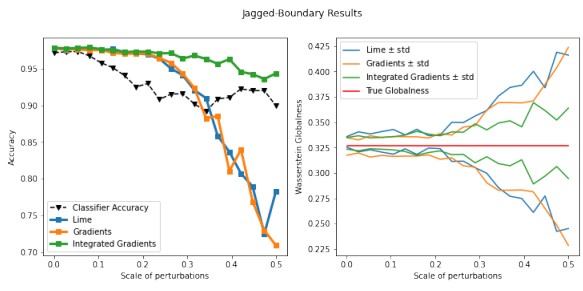

(a) Identifying Grouped Explanations.                    (b) Jagged Boundary Results.

Figure 3: **(a) Identifying Grouped Explanations.** The results of computing Wasserstein Globalness for a variable number of classes, on the MNIST dataset. As we explain an increasing amount of digits, the explanations become more local. **(b) The results of the jagged-boundary experiment.** The accuracy of both the explainers and the model (left) decrease as the Wasserstein Globalness (right) becomes more varied. These results are aggregated across 15 total runs (new data each time). We show the mean $\pm$ one standard deviation of Wasserstein Globalness.

**Explainers.**    Before describing the experiments, we will briefly discuss the explanation methods used in the experiments. The *Gradients* (Simonyan et al., 2013) method directly uses the gradient of class score with respect to the input image as a saliency map. The *Integrated Gradients* method accumulates this gradient at several points along a path from some baseline input to the input to be explained (Sundararajan et al., 2017). The *Expected Gradients* method extends integrated gradients by allowing a distribution over baselines, and computing the expected value of integrated gradients with respect to this distribution (Erion et al., 2021). The *SmoothGrad* method smooths the gradients with a Gaussian kernel to alleviate the high frequency fluctuations commonly seen in partial derivatives (Smilkov et al., 2017). The *Deep-SHAP* method (Lundberg & Lee, 2017) approximates shapley values by extending the *DeepLIFT* algorithm (Shrikumar et al., 2017). *LIME* (Ribeiro et al., 2016) forms a local-approximation to the decision boundary by training a linear model on distance-weighted samples from the input $\mathcal{X}$. *Guided Backpropagation* (Springenberg et al., 2015) modifies the gradient approach by adding a signal that prevents the backwards flow of negative gradients during backpropagation.

**Identifying Grouped Explanations.**    In some problems, one may observe that certain groups of inputs receive similar explanations. In this case, it could be beneficial to determine *how many* such groups exist. Our technique allows us to answer such questions because the number of clusters is inversely related to globalness. In this section, we demonstrate this relationship empirically. Specifically, we observe that the Wasserstein Globalness decreases as we add more classes because the explanations for images in a given class will receive similar explanations. We train a simple MLP classifier on the previously described MNIST dataset, with no regularization and hidden layers of size 300, 500, and 300 respectively. We use cross-entropy loss, and train for 5 epochs.

For all image-data in our experiments, including MNIST, we use a feature attribution explainer. Feature attribution explainers score each feature according to its importance. Because of this, explanations fall in $\mathbb{R}^d$ rather than $\{0,1\}^d$. We again follow the canonical distance metric from Table 1, we will endow our metric space $(\mathcal{E}, d_{\mathcal{E}})$ with the cosine distance: $d_{cos}(x,y) = 1 - \frac{x \cdot y}{\|x\|_2 \|y\|_2}$. Figure 3a shows the results of this experiment. As expected, each explainer correctly decreases its Wasserstein Globalness as we increase the number of digits explained.

**Identifying Local Explainer Stability.**    If explanations change drastically after a very small change to the input, this explainer may be considered less trustworthy because we expect our explainers to be locally-stable. Of course, the degree of this stability depends on the specific problem in question. We will later discuss how this explainer-stability is related to the classifier's stability. By evaluating Wasserstein Globalness in a small neighborhood around a single data point, we can see how robust the explainer is to such perturbations. The following experiments demonstrate this insight.

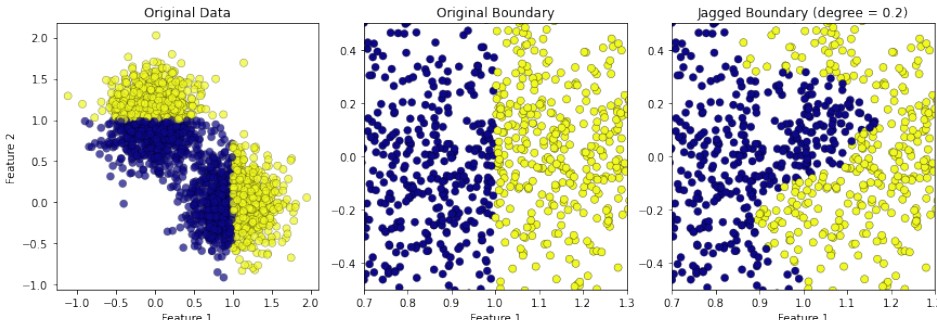

Figure 4: A visualization of the synthetic data in $\mathbb{R}^2$. Data points are illustrated as dots, with color corresponding to class label. Each of the two clusters utilizes a different feature to discriminate between the two classes. The original data is shown (left), alongside a zoomed-in visualization of the original boundary (middle), and a zoomed-in visualization of our jagged-boundary modification (right).

In our first stability experiment, we use the synthetic data shown in Figure 4. We attempt to fool the explainer by making the originally-straight decision boundary jagged, thereby obscuring the true relevant feature. To make this boundary jagged, we randomly visit a point $x$, then flip all nearby points $x'$ to the label of $x$, ensuring that visited points $x$ and $x'$ are never revisited. This data modification is illustrated in Figure 4, and pseudocode is provided in Appendix A, Algorithm 1. In this experiment, we select the feature with the highest attribution, and use the hamming distance in our explanation space: $d_H(x,y) = \sum_i \mathbb{1}(x_i \neq y_i)$. The results in Figure 3b demonstrate how $Acc_{exp}$ is related to the explainer globalness, $G$, in this synthetic-data experiment. We expect that, as explainers become less accurate, the variance of their globalness increases. Accurate explainers, on the other hand, should yield globalness values which are consistently close to the ground-truth globalness.

In Figure 3b, we see the relationship between $Acc_{exp}$ and Wasserstein Globalness. As the explainer accuracy decreases, the Wasserstein Globalness varies farther from the ground-truth Globalness. Furthermore, the integrated gradients method is more robust to the jagged boundaries, and this is reflected in it having the smallest standard deviation of Wasserstein Globalness.

We conduct our next experiment on the CIFAR-10 dataset (Krizhevsky, 2009). This experiment demonstrates how globalness can be used as an indicator of *explainer robustness in a local neighborhood*. First we train a CNN on the dataset as our classification model. We generate a set of inputs by taking one CIFAR-10 image and performing 300 independent adversarial attacks on that image. Each adversarial input is explained, and we evaluate the Wasserstein Globalness for that set of adversarial inputs. This is repeated several times, each time allowing stronger adversarial attacks. Each time we evaluate Wasserstein Globalness we are evaluating it in a larger local neighborhood of $\mathcal{X}$. By doing this, we can examine how globalness differs for different explainers. Though we do not know the ground-truth globalness for these images, we expect the globalness to be maximum in this setting, since feature importances should not be affected by very small perturbations. We are primarily interested in perturbations which are small enough that the classifier still performs well. Intuitively, we hope that, as long as the classification does not change, the *reason* for the classification should also remain roughly constant. Figures 5 and 6 show that, even with only 300 explanations, in $32 \times 32 \times 3$ dimensions, the Wasserstein Globalness can differentiate the explainers.

In Figure 5, we can see how explanations change after the adversarial attacks. This figure also illustrates that the perturbed images are not too different from the original image. Though the images are perceptually near-identical, the explanations can be seen to change, leading to a change in globalness. In Figure 6, we observe exactly how this globalness changes for the batch of explanations at each $\sigma$. The deep-SHAP explainer is the most robust to local perturbations (Wasserstein Globalness close to maximum for a variety of $\sigma$), and that the Gradient and SmoothGrad explainers are the most sensitive. This result is not surprising, as SmoothGrad injects noise in order to perform the smoothing operation, and the Gradient approach is the most naive, forming the basis for many of the other approaches.

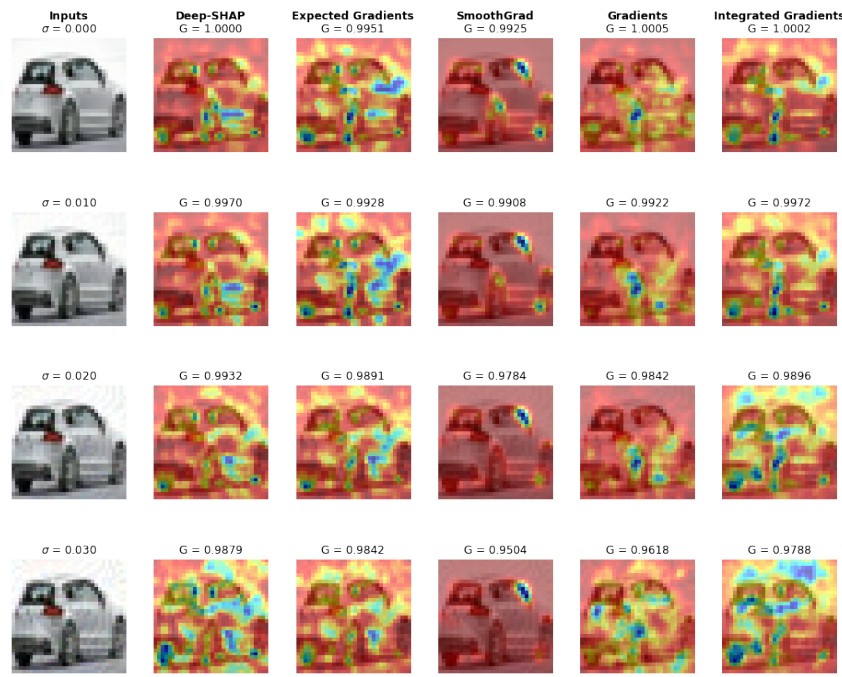

Figure 5: The perturbed input image is shown in the left column. The remaining columns correspond to different explanation methods, and show an explanation overlaid on the original image.

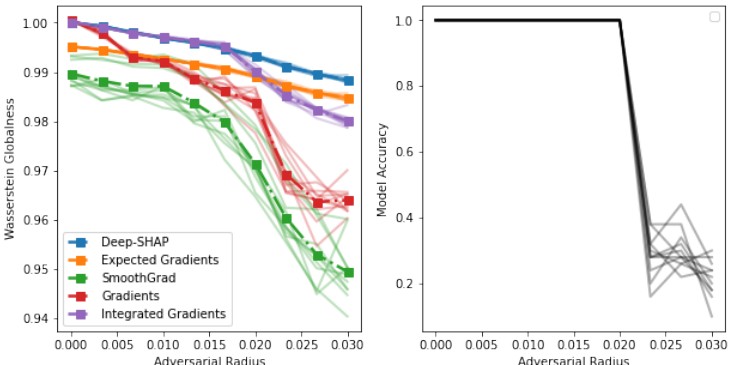

Figure 6: The Wasserstein Globalness for each explainer in our CIFAR-10 experiment (left). The results of 11 separate runs are plotted in the same color for each explainer. The bold line with markers of that color represents the mean across all 11 runs. We also indicate the classifier accuracy (right) in order to indicate the point at which the perturbations become too strong for the explainer.

## 6 CONCLUSION

Explainability is a branch of machine learning that not only facilitates trust between the user and the model, but also allows us to learn from our models. Thus, it is critical that we understand our explainers to use them reliably. In this paper, we have argued that globalness is a theoretical property of explainers which can be valuable to the explainability community. We introduced Wasserstein Globalness, a model-agnostic method for measuring the globalness of explainers which satisfies intuitive axioms regarding the notion of globalness. Finally, we presented experimental results which demonstrate how globalness can be used to differentiate explainers and choose the best one for a given task.

ETHICS STATEMENT

As ML models grow in both complexity and popularity, they have also become less interpretable. These complex models often exhibit surprising behavior. In extreme cases, they can propagate racial and gender bias and even put humans in physical danger. However, even the benign cases erode trust between the user and the model. ML practitioners are responsible for the harmful effects of their models, and are therefore are obliged to build reliable, trustworthy systems.

Explainability is one way in which we can begin to build trustworthy models. It is crucial that we seek a better understanding of uninterpretable models in order to deploy them safely in the real world, whether that means building interpretable models in the first place or explaining them after the fact. In this paper, we discuss a theoretical property of explainability methods, and present a technique for measuring this quantity in practice. We intend for our globalness measure to facilitate responsible practices by enabling humans to better understand their models. This understanding can facilitate responsible decisions, enable practitioners to remove harmful behavior from their systems, and increase our trust in the models themselves.

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

## A   APPENDIX

Table 2 provides a summary of the notation and terminology used in the paper.

| Symbol | Name |
|--------|------|
| $\phi_{\#}$ | Push-forward Measure |
| $\mathbb{1}$ | Indicator Function |
| $\delta_x$ | Dirac distribution (centered at x) |
| $\mathbb{E}$ | Expected Value |
| $\mathcal{E}$ | Set of Explanations |
| $\mathcal{X}$ | Space of model inputs |
| $E$ | Explainer |
| $P(X)$ | Set of probability distributions over $X$ |
| $U$ | Uniform Distribution |
| $\Pi(P, Q)$ | Set of joint distributions which marginalize to $P$ and $Q$ |
| $\mathbb{R}$ | Real numbers |
| $\langle \cdot, \cdot \rangle_F$ | Frobenius Inner Product |
| $d_{\mathcal{E}}$ | distance between explanations |
| mm-space | metric-measure space |

Table 2: Summary of notation used in the paper

Algorithm 1 describes the jagged-boundary data modification shown in Figure 4.

---

**Algorithm 1** Jagged-Boundary

---

1: Let $X$ be a finite data sample.
2: Let $label(\cdot)$ give the label of each point in $X$.
3: Let $N_x$ be a neighborhood around point $x$.
4: **while** not all points visited **do**
5:     **for** unvisited $x$ **do**
6:         **for** unvisited $x' \in N_x$ **do**
7:             $label(x') \leftarrow label(x)$
8:             $x'$ visited
9:         **end for**
10:     **end for**
11: **end while**

---

We measure the wall clock timing of the three different computationally expensive tasks. Because these quantities vary with the explainer used and the random input image, we present the average over both the explainers and 10 randomly selected CIFAR-10 images. These results are shown in Figure 3.

| Task | Average Time (s) |
|------|------------------|
| Adversarial Perturbations | 56.17 |
| Generating Explanations | 19.42 |
| Computing Globalness | 6.97 |

Table 3: Average wall clock time for the CIFAR-10 experiment. As expected, computing globalness is less computationally expensive than generating the explanations themselves.

Figure 7 shows all of the perturbed images from the CIFAR-10 experiment presented in 5. Figure 5 is formed from a subset of the rows in Figure 7. Figure 8 depicts another set of explanations, with a different original input image.

In figure 11, we show an example of the different perturbed images for a single adversarial radius $\sigma$.

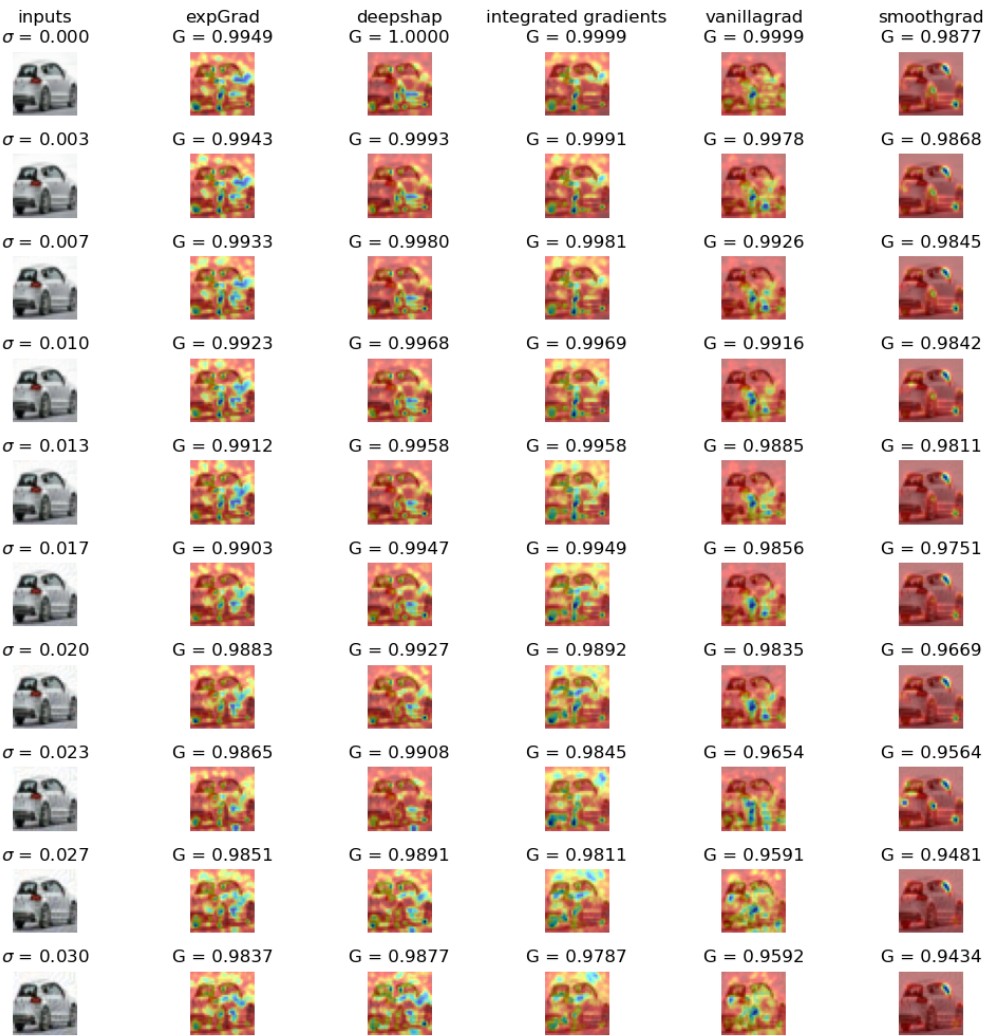

Figure 7: An extended version of 5

# B   APPROXIMATING WASSERSTEIN GLOBALNESS IN HIGH-DIMENSIONS

Computing the Wasserstein distance traditionally involves solving a linear programming problem which does not scale well. In the previous section, we discussed the sample complexity of $\hat{G}_p(\mu_N)$. This empirical estimate also scales poorly with the dimension of the data. However, several techniques have been developed to efficiently compute Wasserstein distances in high dimensions. One such technique, called entropy-regularized optimal transport, relies on regularizing the entropy of the joint distribution $\pi \in \Pi(P, Q)$.

The Wasserstein distance can be written as an optimization over the set of product measures $\Pi(P, Q)$. In its discrete form, the expectation $\mathbb{E}_{(x,y)\sim\pi}[d_{\mathcal{E}}(x, y)]$ from Equation 7 can be written as a Frobenius inner product between the matrix of distances $D$, and the transport plan $R$.

$$d_W^1(p, q) = \min_{R \in \Pi(p,q)} \langle R, D \rangle_F \tag{11}$$

The sinkhorn distance, introduced to the ML community by Cuturi (2013), utilizes a technique called entropic regularization. The entropy-regularized transport problem adds an additional regularization term $\Omega(R) = \sum_{i,j} r_{i,j} log(r_{i,j})$.

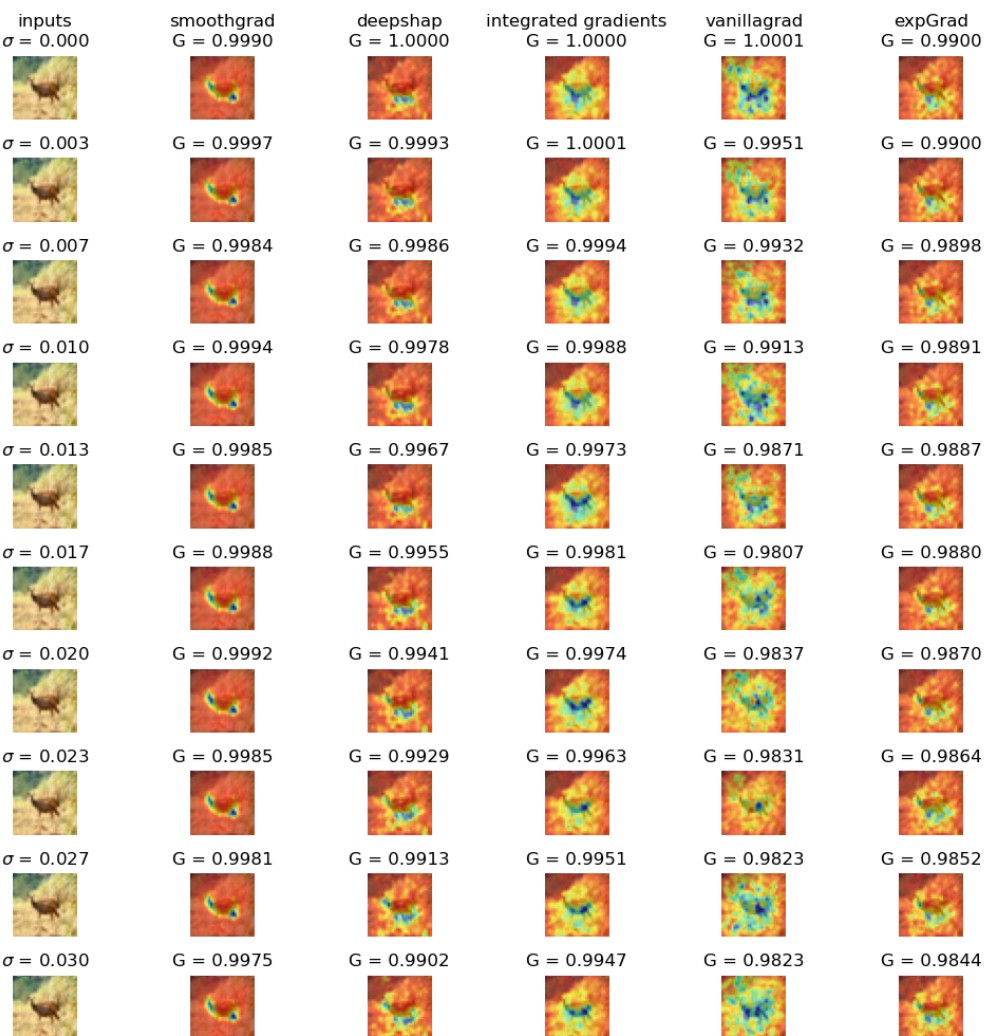

Figure 8: Another set of explanations for an image of a deer from the CIFAR-10 dataset.

$$d_{sinkhorn}(p, q) = \min_{R \in \Pi(p,q)} \langle R, D \rangle_F + \lambda \Omega(R) \tag{12}$$

This encourages the transport plan to have less mutual information with the marginals $p$ and $q$. By searching just over these "smooth" transport plans, we have convexified the optimization problem, and now can use matrix scaling algorithms to solve it [1]. There are also existing theoretical guarantees that $d_{sinkhorn}$ will be close to $d_W$, for which we refer the reader to Cuturi (2013).

By utilizing entropic regularization, we can find good solutions faster. However, we can always decrease the strength of this regularization in order to eliminate noise from the computation. As we have discussed, this is especially useful when we work with high-dimensional data, as is done in 5.

## C   PROOF OF THEOREM 1

We will now restate and prove that the axioms from Section 3.2 apply to the definition of $G$ in Section 3.4.

---

[1]For our experiments, we compute the sinkhorn algorithm using available optimal transport solvers available at Flamary et al. (2021).

Figure 9: Another set of explanations for an image of a ship from the CIFAR-10 Dataset.

**Property 1** states that $G_p(\mu) \geq 0 \; \forall \mu$.

*Proof.* Since $d_W$ is a distance metric, it is non-negative by construction. Therefore $G_p(P) = d_W(P, U)$ is non-negative. $\qquad\square$

**Property 4** Let $\{\mu_n\}$ be a sequence of probability distributions which converges in distribution to $\mu$. Then $G_p(\mu_n) \to G_p(\mu)$.

*Proof.*

By remark 2.4 in (Mémoli, 2011),

$$|d_W^p(A, B) - d_W^p(A_n, B_n)| \leq d_W^p(A, A_n) + d_W^p(B, B_n) \tag{13}$$

This implies that

$$|d_W^p(\mu, U) - d_W^p(\mu_n, U_n)| \leq d_W^p(\mu, \mu_n) + d_W^p(U, U_n) \tag{14}$$

However, $U$ does not need to be approximated, therefore

$$U = U_n \implies |G_p(\mu) - G_p(\mu_n)| = |d_W^p(\mu, U) - d_W^p(\mu_n, U)|$$
$$\leq d_W^p(\mu, \mu_n)$$

| inputs
$\sigma = 0.000$ | smoothgrad
G = 0.9992 | deepshap
G = 1.0000 | integrated gradients
G = 1.0001 | vanillagrad
G = 0.9996 | expGrad
G = 0.9925 |
|---|---|---|---|---|---|
| $\sigma = 0.003$ | G = 0.9991 | G = 0.9980 | G = 0.9994 | G = 0.9962 | G = 0.9917 |
| $\sigma = 0.007$ | G = 0.9993 | G = 0.9960 | G = 0.9978 | G = 0.9938 | G = 0.9907 |
| $\sigma = 0.010$ | G = 0.9993 | G = 0.9948 | G = 0.9977 | G = 0.9919 | G = 0.9899 |
| $\sigma = 0.013$ | G = 0.9984 | G = 0.9933 | G = 0.9967 | G = 0.9895 | G = 0.9889 |
| $\sigma = 0.017$ | G = 0.9990 | G = 0.9915 | G = 0.9953 | G = 0.9875 | G = 0.9877 |
| $\sigma = 0.020$ | G = 0.9990 | G = 0.9896 | G = 0.9953 | G = 0.9855 | G = 0.9857 |
| $\sigma = 0.023$ | G = 0.9983 | G = 0.9879 | G = 0.9938 | G = 0.9833 | G = 0.9843 |
| $\sigma = 0.027$ | G = 0.9969 | G = 0.9859 | G = 0.9932 | G = 0.9814 | G = 0.9831 |
| $\sigma = 0.030$ | G = 0.9971 | G = 0.9850 | G = 0.9936 | G = 0.9807 | G = 0.9823 |

Figure 10: Another set of explanations for an image of a truck from the CIFAR-10 dataset.

$\sigma = 0.03$

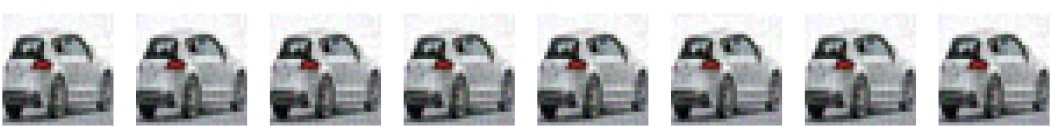

Figure 11: A visualization of the pertubed images used in our CIFAR-10 experiment. These correspond to the last row in Figure 7.

But $\mu_n$ converges in distribution to $\mu$, meaning $d_W^p(\mu, \mu_n) \to 0 \implies |G_p(\mu) - G_p(\mu_n)| \to 0$ $\quad\square$

From this proof we can see that, not only does $G_p(\mu_n)$ converge to $G_p(\mu)$, but it converges at least as quickly as $\mu_n \to \mu$.

**Property 3** states that there exists an $x_0 \in \mathcal{E}$ for which $G_p(\delta_{x_0}) \geq G_p(\mu) \ \forall \mu \in \mathcal{P}(\mathcal{E})$. Furthermore, we will also show that, for this $x_0$, $s_{x_0,p} \leq s_{x,p} \ \forall x \in \mathcal{E}$, where $s_{x,p}$ is the *p-eccentricity*, defined in Mémoli (2011) as $s_{x,p} = \int_X d^p(x, x') dx'$ [2].

---

[2]Roughly, the p-eccentricity indicates how far a point $x$ is to the rest of the points in $\mathcal{E}$.

P-eccentricity appears in this property in order to characterize the $x_0$ for which $\delta_{x_0}$ achieves maximum globalness.

*Proof.* . Without loss of generality, assume $s_{x_0,p} \geq s_{x,p}, \ \forall x \in \mathcal{E}$. Any **discrete** distribution can be represented as a delta-train $P = \sum_i \lambda_i \delta(x_i)$ such that $\lambda_i \geq 0, \ \forall i$ and $\sum_i \lambda_i = 1$. By Property 6, we know that $G_p(P) \leq \sum_i \lambda_i G_p(\delta(x_i))$. But for a Dirac measure, 7 reduces to

$$G_p(\delta_{x_0}) = \Big( \int_X d^p(x_0, x) dx \Big)^{1/p} = s_{x_0,p} \tag{15}$$

So

$$G_p(P) \leq \sum_i \lambda_i s_{x_0,p} \leq \max_i s_{x_i,p} = s_{x_0,p}. \tag{16}$$

This proves the axiom for any discrete distribution. But general distributions can be approximated with these weighted delta-trains, so by Property 4, so in the limit, this proof holds for general distributions. $\qquad\square$

**Property 2** states that $\mu = U \iff G(\mu) = 0$

*Proof.* First note that $d_W^p(P, Q)$ is a metric. From the identity of indiscernibles axiom of distance metrics, $(x, y) = 0 \iff x = y$. This directly implies that $d_W(\mu, U) = 0 \iff \mu = U$. $\qquad\square$

We will now prove the **first part of Property 5**, restated below:

Let $T_{\mathcal{E},d_\mathcal{E}}$ be the group of isometries of $(\mathcal{E}, d_\mathcal{E})$ defined in Equation 2. Then $G_p$ is $T_{\mathcal{E},d}$-invariant.

*Proof.* We need to show that $G_p(\mu) = G_p(\phi_\# \mu), \ \forall \phi \in T_{\mathcal{E},d}$.

One consequence of the definition of push-forward measure is that

$$P_{\phi_\# q}(\phi x) = P_q(\phi^{-1} \phi x) = P_q(x) \implies \mathop{\mathbb{E}}_{x \sim \phi_\# q}[\phi x] = \mathop{\mathbb{E}}_{x \sim q}[x] \tag{17}$$

Consider the set of measure couplings (feasible transport plans) between $\mu$ and $v$, denoted $\Pi(\mu, v)$

Let $h_\phi : \Pi(\mu, v) \to \Pi(\phi_\# \mu, \phi_\# v)$ be given by

$$h_\phi \circ \pi(A, B) = \pi(\phi^{-1} A, \phi^{-1} B) \tag{18}$$

Since $h_\phi$ is bijective, and since we have shown in 17 that $\mathbb{E}_{x \sim \pi}[x] = \mathbb{E}_{x \sim h_\phi(\pi)}[\phi(x)]$, it holds that

$$G_p(\mu) = \inf_{\pi \in \Pi(\mu,U)} \mathbb{E}_{(x,y) \sim \pi}[d^p(x, y)]^{1/p} \tag{19}$$

Because $\phi$ is an isometry,

$$= \inf_{\pi \in \Pi(\mu,U)} \mathbb{E}_{(x,y) \sim h_\phi(\pi)}[d^p(\phi x, \phi y)]^{1/p} \tag{20}$$

Because $\mathbb{E}_{x \sim \pi}[x] = \mathbb{E}_{x \sim h_\phi(\pi)}[\phi(x)]$,

$$= \inf_{\pi \in \Pi(\mu,U)} \mathbb{E}_{(x,y) \sim h_\phi(\pi)}[d^p(x, y)]^{1/p} \tag{21}$$

Finally, since $h_\phi$ is a bijection between $\Pi(\mu, U)$ and $\Pi(\phi_\# \mu, \phi_\# U)$,

$$= \inf_{\pi \in \Pi(\phi_\# \mu, \phi_\# U)} \mathbb{E}_{(x,y) \sim \pi}[d^p(x, y)]^{1/p}$$
$$= G_p(\phi_\# \mu)$$

$\qquad\square$

We will now prove the **second part of Property 5**, which states that $\exists \phi : \mathcal{E} \to \mathcal{E} \notin T_{\mathcal{E},d_\mathcal{E}}$ for which $G_p(\phi_\# \mu) \neq G_p(\mu)$, for some $\mu$.

*Proof.* Consider a $\phi \in S$, $\phi : \mathcal{E} \to \mathcal{E}$ such that $d_\mathcal{E}(x,y) = c * d_\mathcal{E}(\phi x, \phi y)$ for some constant $c$, and some $x, y \in \mathcal{E}$. Note that $\phi$ is clearly not in $T_{\mathcal{E},d_\mathcal{E}}$, the group of isometries of $(\mathcal{E}, d_\mathcal{E})$.

Like before, we can write:

$$G_p(\mu) = \inf_{\pi \in \Pi(\mu,U)} \mathbb{E}_{(x,y)\sim\pi}[d^p(x,y)]^{1/p} \tag{22}$$

$$= \inf_{\pi \in \Pi(\phi_\#\mu,\phi_\#U)} \mathbb{E}_{(x,y)\sim\pi}[d^p(x,y)]^{1/p} \tag{23}$$

But since $\phi$ is not isometric:

$$= \inf_{\pi \in \Pi(\phi_\#\mu,\phi_\#U)} \mathbb{E}_{(x,y)\sim\pi}[c^p d^p(x,y)]^{1/p} \tag{24}$$

$$\neq G_p(\phi_\#\mu) \tag{25}$$

$\square$

**Property 6** (convexity) Let $R$ be a convex combination of measures $P$ and $Q$. That is, $R = \lambda P + (1-\lambda)Q$ for some $\lambda \in [0,1]$. Then $G(R) \leq \lambda G(P) + (1-\lambda)G(Q)$.

*Proof.* Let $P, Q$ be probability distributions. Let $\phi_P$ be the optimal transport plan between $P$ and $U$. Similarly, let $\phi_Q$ be the optimal transport plan between $Q$ and $U$. Then the convex combination of $P$ and $Q$ is a feasible transport plan between $R = \lambda P + (1-\lambda)Q$ and $U$, so

$$G_p(X) = \inf_{\pi \in \Pi(\mu,U)} \mathbb{E}_{(x,y)\sim\pi}[d^p(x,y)]^{1/p} \tag{26}$$

$$\leq \mathbb{E}_{(x,y)\sim\lambda\phi_P + (1-\lambda)\phi_Q}[d^p(x,y)]^{1/p} \tag{27}$$

$$= \lambda \mathbb{E}_{(x,y)\sim\phi_P}[d^p(x,y)]^{1/p} + (1-\lambda)\mathbb{E}_{(x,y)\sim\phi_Q}[d^p(x,y)]^{1/p} \tag{28}$$

$$= \lambda G_p(P) + (1-\lambda)G_p(Q) \tag{29}$$

$\square$

# D    PROOF OF THEOREM 2

We will first introduce two lemmas to simplify the proof of theorem 2. The first lemma concerns the error incurred by approximating $\mu$ with finite samples. The second lemma concerns the error incurred by approximating $U$ with finite samples. By bounding both of these errors, we obtain an upper bound on the error of using both finite-sample approximations.

**Lemma 1.** *Let $p \in (0, d/2)$. Also, let $q \neq d/(d-p)$. Let $M_q(\mu) = \int_{\mathbb{R}^d} |x|^q d\mu(x)$. Then for some constant $C_{p,q,d}$ which depends only on p,q, and d,*

$$\mathbb{E}\big[|G_p(\mu) - G_p(\mu_N)|\big] \leq C_{p,q,d} M_q^{p/q}(\mu)(N^{-p/d} + N^{-(q-p)/q}) \tag{30}$$

*Proof.* From the triangle inequality,

$$d_W^p(\mu_N, U) \leq d_W^p(\mu, U) + d_W^p(\mu_N, \mu) \tag{31}$$

From (Fournier & Guillin, 2015), $\mathbb{E}[d_W^p(\mu_N, \mu)] \leq C_{p,q,d} M_q^{p/q}(\mu)(N^{-p/d} + N^{-(q-p)/q})$.

So

$$\mathbb{E}[G_p(\mu_N)] - G_p(\mu) \leq C_{p,q,d} M_q^{p/q}(\mu)(N^{-p/d} + N^{-(q-p)/q}) \tag{32}$$

Similarly, we can invoke the triangle inequality with $d_W^p(\mu, U)$ on the LHS:

$$d_W^p(\mu, U) \leq d_W^p(\mu_N, U) + d_W^p(\mu_N, \mu) \tag{33}$$

And, making the same argument, we get:

So

$$G_p(\mu) - \mathbb{E}[G_p(\mu_N)] \leq C_{p,q,d} M_q^{p/q}(\mu)(N^{-p/d} + N^{-(q-p)/q}) \tag{34}$$

Finally, combining 32 and 34,

$$\mathbb{E}\big[|G_p(\mu) - G_p(\mu_N)|\big] \leq C_{p,q,d} M_q^{p/q}(\mu)(N^{-p/d} + N^{-(q-p)/q})$$

$\square$

This theorem is related to Property 4 in the sense that the empirical distribution $\mu_N$ converges to $\mu$. This theorem is concerned with the *rate* at which $G_p(\mu_N)$ converges, whereas Property 4 simply states that it will converge.

Perhaps the most important consequence of 1 is that $\mathbb{E}\big[|G_p(\mu) - G_p(\mu_N)|\big] \to 0$ as $N \to \infty$.

**Lemma 2.** *Let $p \in (0, \frac{d}{2})$, and $q > \frac{dp}{d-p}$. Then for some constant $\kappa_{p,q}$ which depends only on $p$ and $q$,*

$$\mathbb{E}\big[|\hat{G}_p(\mu_N) - G_p(\mu_N)|\big] \leq \kappa_{p,q} \Big[ \int_{\mathbb{R}^d} \|x\|^q d\mu(x)\Big]^{1/q} N^{-1/d} \tag{35}$$

*Proof.* From the triangle inequality:

$$d_W(\mu_N, U_N) \leq d_W(\mu_N, U) + d_W(U_N, U) \tag{36}$$
$$\implies d_W(\mu_N, U_N) - d_W(\mu_N, U) \leq d_W(U_N, U) \tag{37}$$

From (Dereich et al., 2013), there exists some constant $\kappa_{p,q}$ such that $\mathbb{E}[d_W^p(U_N, U)] \leq \kappa_{p,q}\big[\int_{\mathbb{R}^d} \|x\|^q d\mu(x)\big]^{1/q} N^{-1/d}$.

$$\implies \mathbb{E}[d_W(\mu_N, U_N) - d_W(\mu_N, U)] \leq \mathbb{E}[d_W^p(U_N, U)] \leq \kappa_{p,q}\big[\int_{\mathbb{R}^d} \|x\|^q d\mu(x)\big]^{1/q} N^{-1/d}$$

$$\implies \mathbb{E}[\hat{G}_p(\mu_N) - G_p(\mu_N)] \leq \kappa_{p,q}\big[\int_{\mathbb{R}^d} \|x\|^q d\mu(x)\big]^{1/q} N^{-1/d}$$

Similarly:

$$d_W(\mu_N, U) \leq d_W(\mu_N, U_N) + d_W(U_N, U)$$
$$\implies d_W(\mu_N, U) - d_W(\mu_N, U_N) \leq d_W(U_N, U)$$

So we have that:

$$\mathbb{E}[|\hat{G}_p(\mu_N) - G_p(\mu_N)|] \leq \kappa_{p,q}\Big[\int_{\mathbb{R}^d} \|x\|^q d\mu(x)\Big]^{1/q} N^{-1/d} \tag{38}$$

$\square$

We will now combine the previous lemmas to get a final bound on the approximation error incurred by using discrete approximations for both $\mu$ and $U$.

*Proof.* The following one-line proof is a natural consequence of lemmas 1 and 2.

$$\mathbb{E}[|\hat{G}_p(\mu_N) - G_p(\mu)|] \leq \mathbb{E}[|\hat{G}_p(\mu_N) - G_p(\mu_N)|] + \mathbb{E}\big[|G_p(\mu) - G_p(\mu_N)|\big]$$

$$\leq \kappa_{p,q}\Big[\int_{\mathbb{R}^d} \|x\|^q d\mu(x)\Big]^{1/q} N^{-1/d} + C_{p,q,d} M_q^{p/q}(\mu)(N^{-p/d} + N^{-(q-p)/q})$$

$$= \mathbb{E}[|\hat{G}_p(\mu_N) - G_p(\mu)|] \leq \kappa_{p,q} M_q(\mu)^{1/q} N^{-1/d} + C_{p,q,d} M_q^{p/q}(\mu)(N^{-p/d} + N^{-(q-p)/q})$$

$\square$

# E    PROOF OF REMARK 3.3

*Proof.* Let $\psi \in S$ be a permutation of the set of explanations.

By Equation 17, a direct consequence of the definition of push-forward measure, we have that:

$$\mathbb{E}_{x\sim\mu}[g(\mu(x))] = \mathbb{E}_{x\sim\psi_{\#}\mu}[g(\psi_{\#}\mu(x))] \tag{39}$$

for some function $g$.

Now observe that both entropy and f-divergences, as defined in Equations 5 and 6 respectively, can be written in the form above. Entropy is obtained by directly letting $g(x) = -\log(x)$.

$$H(X) = \mathbb{E}_{x\sim\mu}[-\log(\mu(x))] \tag{40}$$

The f-divergence between $\mu$ and $U$ is equivalent by allowing $g(x) = f(\frac{c}{x})$, and allowing the constant $c$ to be the uniform probability of an element $x$, $U(x)$.

$$D(U\|\mu) = \mathbb{E}_{x\sim\mu}\left[f(\frac{U(x)}{\mu(x)})\right] \tag{41}$$

$\square$

