# OpenReview forum: "Axiomatic Explainer Locality With Optimal Transport"
_ICLR.cc/2023/Conference — Submitted to ICLR 2023_

### Official Review · Reviewer_CcYU · 2022-10-24

**Confidence:** 3
**Correctness:** 3
**Technical Novelty And Significance:** 3
**Empirical Novelty And Significance:** 2
**Recommendation:** 3

**Clarity, Quality, Novelty And Reproducibility:**

This submission has a list of writing issues that make it hard to understand.

1.  "Let $\mu \in P(\mathcal{E})$ be one such probability distribution". Where is this distribution from? What's a random event? It seems that the distribution is derived from the model and the data, but you may want to make it clear so the reader does not have to guess.

2. $\mu$ is used a second time in "$\mu \in P(X \times Y)$".

3. The reading of Property 5 is bumpy. You may want to move the definition of S to the sentence before (4). Additionally, I don't understand why we need to consider G is NOT invariant to S?

4. In property 6, what do you mean by "combining" two explanations?

5. The notation in 3.4 is problematic: $\mathcal{E}$ is the space of explanations, but the discussion seems to define a metric for $P(\mathcal{E})$, the distribution space of explanations.

6. I'd like to see a clear statement of the goal of the experiment section at the beginning of section 5.


**Details Of Ethics Concerns:**

No concern.

**Strength And Weaknesses:**

Strength:

+ The work considers globalness of explanations from a theoretical perspective. I don't know this area well, but there is no such theoretical study before, I feel this contribution is solid.
+ The work analyzes a list of properties that a metric of globalness should satisfy.

Weaknesses:

- The analysis is somewhat shallow. I feel that most analyses are not far from my intuitive understanding.
- The proposed approach of Wasserstein Globalness is not examined against those properties analyzed in the first part of the work.
- The writing of the submission is poor. It is hard to understand,  partially because it contains many typos.

Besides these weaknesses, I don't see a clear goal of experiments.


**Summary Of The Paper:**

The paper studies the problem of measuring globalness of neural network explainers. The work considers the globalness of explanations from a theoretical angle and inspected a list of properties that a metric of globalness should satisfy.

**Summary Of The Review:**

Overall, this work studies the interesting problem of measuring globalness of explanations in a more rigorous way. However, there is a disjoint between the analysis and the proposed approach. It is also unclear the benefit of the this new metric of globalness. The writing of the work should be improved.

---

> ### Author Response · Authors · 2022-11-13
> **Response to Reviewer CcYU**
>
> Thank you for your constructive comments, and for highlighting that this is the first work that formally defines globalness in a theoretical way.
>
> We realized that by your second weakness bullet point (“Wasserstein Globalness is not examined against those properties analyzed in the first part of the work”) that you may have missed Theorem 1 in our paper.  Theorem 1 states that Wasserstein Globalness satisfies these properties.  Furthermore, in the appendix we provide the full proof of this theorem, demonstrating that each listed property is satisfied by Wasserstein Globalness.  Please take a second look at our paper and note that we have the necessary theoretical rigor.
>
>
> We would like to briefly point out that, in your summary, you suggest that our method is limited to neural network explainers.  This is not true; our analysis is model agnostic and is very general. It can be applied to explainers for any machine learning model.
>
>
> We will now address the remaining bulleted weaknesses:
> * Regarding the depth of our analysis, we intentionally presented the properties in section 3 such that they would align with your intuition. We did this in order to motivate our measure from an axiomatic perspective, which must necessarily be close to your intuitive understanding. Again we would like to direct your attention to the supplementary material, where we include a rigorous proof for each property.
> * We have made a second-pass and corrected any typos found.
>
> Thank you for the minor clarity suggestions; we have revised our paper accordingly.
>
> 1. You are correct that this distribution is generated by passing data through our explainer. We have clarified this in the first paragraph of section 3.1.
>
> 2. We have removed this ambiguity.
>
> 3. We agree with this suggestion and have modified property 5 accordingly. We include this negative statement about S because we want a measure of globalness to capture the geometry of explanations. If the measure was invariant to S, then it would necessarily ignore the geometry of explanations.
>
> 4. We mean something like this: consider you have two explainers E_1 and E_2. When you see an input, generate an explanation from E_1 with probability p, or an explanation from E_2 with probability (1-p). We have added this clarification under Property 6 on page 4.
>
> 5. If we understand your concern correctly, you may be confusing two distance metrics. We define d_{\Epsilon} as the distance between explanations. We also consider the Wasserstein distance, which utilizes this distance between explanations to measure a distance between distributions of explanations.
>
> 6. We have clarified the introductory paragraph of our experiments section. We also would like to direct your attention to the boldface paragraph headers “Identifying Grouped Explanations” and “Identifying Local Explainer Stability”, which name our two flavors of experiments and describe their purpose.

---

### Official Review · Reviewer_8L8j · 2022-10-26

**Confidence:** 2
**Correctness:** 3
**Technical Novelty And Significance:** 3
**Empirical Novelty And Significance:** 3
**Recommendation:** 5

**Clarity, Quality, Novelty And Reproducibility:**

The paper is rather clear even for a non-expert. However, I found it hard to assess the novelty/originality of the work. This is reflected in my confidence score. The authors have not shared code for their experiments, but provide details for replicating the methods/experiments.


**Strength And Weaknesses:**

*Strengths*

* The idea becomes rather clear even for a non-expert, and the paper could be of interest to the explainability community.

* The paper reads well and is easy to follow.

*Weaknesses*

* As a non-expert in explainability, I found it hard to position the work or assess the novelty/originality of this paper.

* The experiment are rather simple, which is good for illustrative purposes, but leaves open questions regarding relevance and generality of the analysis.

**Summary Of The Paper:**

The paper introduces Wasserstein Globalness, a model-agnostic method for measuring the globalness (the inverse of locality) of explainers.


**Summary Of The Review:**

A potentially interesting paper that introduces a concept and evaluates it on simple test scenarios.

---

> ### Author Response · Authors · 2022-11-13
> **Response to Reviewer 8L8j**
>
> Thank you for your comments. Regarding reproducibility, we are sharing anonymous code by which our experiments can be replicated.
>
> Our work presents the first existing measure of explainer locality/globalness. Our paper is novel and original in this sense.
>
> We believe that our experiments are not excessively simple, but instead connect the property of globalness to interesting and meaningful concepts in explainability, such as local adversarial robustness. Locality/globalness is a topic that is relevant to the explainability community. Our analysis has been made as general as possible, as it is model-agnostic and applicable to any type of explainer (feature attribution, feature selection, etc.). We have even left the order of Wasserstein distance general in all of our analyses. We are happy to further discuss any concerns regarding the relevance or generality of our approach.

---

### Official Review · Reviewer_arKF · 2022-10-30

**Confidence:** 5
**Correctness:** 3
**Technical Novelty And Significance:** 3
**Empirical Novelty And Significance:** 2
**Recommendation:** 6

**Clarity, Quality, Novelty And Reproducibility:**

The paper is clearly written and is a quality work. The code is currently not available, but proposed for release. Thus it is not easy, if at all possible, to reproduce the results at the review stage.

As for novelty, I have some reservations. The abstract claims introduction of a novel measure of globalness, yet I've come across this concept already in the paper below. Admittedly the method in that paper was developed to demonstrate an application. This work may be a mathematical treatment of that method. It would be good to have an explanation and discussion of this.

Md Rahman, et al. Interpreting models interpreting brain dynamics. Scientific Reports, 12(1):1–15, 202

**Strength And Weaknesses:**

# Strengths

1.  Clear motivation of the chosen measure with the 5 properties.
2.  An important problem the paper aims to address.
3.  Empirical demonstration with some theoretical justification, and the code provided.
4.  A well written manuscript.

# Weaknesses

1.  Some claims do not take into account the existing literature.
    1.  In the intro it is claimed that there are no obvious metrics to compare the explainers, yet the field of developing such metrics is a highly active one.
        1.  Remove and Retrain ROAR: Sara Hooker et al. A benchmark for interpretability methods in deep neural networks. In Advances in Neural Information Processing Systems, pages 9737–9748, 2019
        2.  Retain and Retrain RAR: Md Rahman, et al. Interpreting models interpreting brain dynamics. Scientific Reports, 12(1):1–15, 202
        3.  Accuracy Information Curves, Softmax Information Curves: Andrei Kapishnikov, Subhashini Venugopalan, Besim Avci, Ben Wedin, Michael Terry, and Tolga Bolukbasi. Guided integrated gradients: An adaptive path method for removing noise. In Proceedings of the IEEE/CVF Conference on Computer Vision and Pattern Recognition, pages 5050–5058, 202
    2.  The abstract claims introduction of a novel measure of globalness, yet I've come across this concept already in the paper below. Admittedly the method in that paper was developed to demonstrate an application. This work may be a mathematical treatment of that method. Please explain.
        1.  Md Rahman, et al. Interpreting models interpreting brain dynamics. Scientific Reports, 12(1):1–15, 202
2.  Judging by experiments, sensitivity of the Wasserstein measure to globalness is low.
    1.  In all experiments the changes of the measure are within hundredth units, while the variance, where shows is much larder.
    2.  In Figure 6 accuracy drops to random, while the measure only slightly decreases and even then not for all methods.
    3.  What is the source of variability in assessing the distribution of explainer predictions? Multiple classifiers each trained with random seeds and explained at each sample? Multiple samples on the same classifier? Something else?
3.  Lack of demonstration of the computational complexity and wall clock timing of the method.
    1.  Wasserstein distance is computationally demanding to compute, especially in high dimensions. There's a danger, that the proposed method will be intracktable in practice despite for toy problems. The paper would benefit from demonstrations on the issue.
4.  The meaning of the distance of the value of it in separating one method from another in terms of behavior of explaners at individual input samples is not clearly demonstrated.
    1.  It is unclear what does it mean for the value to be 0.99 or 0.98, what changes in detected explainers? Figure 5 contains examples of saliency maps, yet interpretation or connection of the proposed G measure with what is displayed is not clear.
    2.  It is clear, that if the metric is close to 0 all saliency maps for each sample must have very few things in common, and a high value of the metric means there is a single global per class feature. Rigorous demonstration that this is indeed the case would help evaluate the value of the proposed method.

**Summary Of The Paper:**

The paper promotes the use of the Wasserstein distance between the distribution of explanations produced by an explainer algorithm and the uniform distribution as a way to access whether the explainer is local (provides sample specific information) or global (class specific). Five desired properties are introduced to motivate the choice, and proofs are provided that the proposed use of the Wasserstein distance concurs with these properties.

**Summary Of The Review:**

A clear paper with a solid and reasonably well-presented idea on an important topic. However, it lacks some demonstration of the claims and does not systematically explore the usefulness of the proposed method.

---

> ### Author Response · Authors · 2022-11-13
> **Response to Reviewer arKF**
>
> Thank you for your thoughtful and detailed review.
>
> 1.1) Finding metrics by which we can compare explainers is an active area of research, and as discussed in our related works, there are several existing post-hoc metrics for evaluating an explainer, like ROAR (Sara Hooker et al) or the energy-based pointing game proposed by (Wang et. al).
>
> However we would like to highlight that our method introduces a novel measure which allows us to compare explainers beyond the post-hoc performance measures discussed here and in our related works section. Unlike previous work, this work explores the concept of locality. This is an important property once we begin to investigate other explainer properties which are related to globalness, like explainer robustness. Local explainer robustness is very important for trustworthy explainers. Wasserstein Globalness can allow us to compare the local robustness of different explainers, as illustrated in our experiments (Figure 6).
>
> Furthermore, there are limitations of existing methods for comparing explainers. For example, the energy-based pointing game is limited to saliency methods for computer vision. ROAR requires a model to be retrained several times, which is not only computationally expensive but can also produce different models from the same data (unstable). Our method, on the other hand, is model-agnostic.
>
> 1.2)  The paper referred to, Rahman et al., used earth-movers (Wasserstein order-1) distance in a standard way. Wasserstein distances are commonly used in applications, and are not new by any means, having been originally defined in 1939: Kantorovich, Leonid V. "The mathematical method of production planning and organization." Management Science 6.4 (1939): 363-422. Unlike Rahman et al., we investigate a property of explainers, locality, theoretically, suggest a way to measure this property, and prove that this measure satisfies all desired axioms of this property; hence, the Rahman et al. paper does not affect the novelty of our work.
>
> Our work is the first paper to study the globalness of explainers from a theoretical angle.
>
> 2.1) In high dimensions, the scale does become small. While we have considered using some normalization to mitigate this, the globalness of explainers will only be compared when those explainers output explanations of the same dimension. Therefore, the relative ordering is what really matters here.
>
> 2.2) You may be confused about the presence of classifier accuracy in this figure. We are not claiming that the globalness measure correlates with classifier accuracy. We show classifier accuracy only to indicate at which point the perturbations become too severe for the classifier, as the classifier’s closest decision boundary affects the locality scale at which we expect adversarial robustness.
>
> To reiterate, this experiment demonstrates that our measure can differentiate the globalness/locality between several explainers, and that this globalness can indicate the local adversarial robustness of the explainer. You say that the globalness drops “even then not for all methods”, but that is the intention of the experiment. The explainers naturally have differing degrees of local adversarial robustness, and we are demonstrating that we can observe this by computing Wasserstein Globalness.
>
> 2.3) In both Figure 3(b) and Figure 6, the variability across runs is coming from random perturbations to the input data.
>
> 3.) We have added results on the wall clock timing in Table 3 in the appendix. We find that computing globalness is the least computationally expensive step in our CIFAR-10 experiment. We would also like to direct you to our Theorem 2, where we present a bound on the sample complexity of Wasserstein Globalness.
>
> 4.1) We have added an additional figure (Figure 11, Appendix) which shows the “hidden” dimension of Figure 5. It is true that the displayed results do not show variance among the saliency maps for each $\sigma$. However, we include figure 5 in order to demonstrate how the saliency maps change as the degree of perturbations is increased.
>
> 4.2) As we point out in our paper, it is well known that Wasserstein distance is a proper distance metric on probability distributions: Clement, Philippe, and Wolfgang Desch. "An elementary proof of the triangle inequality for the Wasserstein metric." Proceedings of the American Mathematical Society 136.1 (2008): 333-339. Because this is a distance metric, values close to 0 indicate that the distribution of explanations is near-uniform, and therefore all saliency maps for each sample must have few things in common. Similarly, a high value of the metric means there is a single global per class feature. These observations are also proved in the paper through our properties 2 and 3.

---

### Author Response · Authors · 2022-11-13
**General Response**

In this work, we study the property of explainer globalness in a theoretical way. While there are existing benchmark methods that attempt to quantify the accuracy of an explainer to true explanations, explainers have other properties beyond accuracy that are critical to understand if we want to deploy trustworthy explainability methods in practice. For example, robustness is one such property that is commonly studied in both explainers and prediction models. Globalness is another property of interest. As the first work to introduce a measure of explainer globalness/locality, we introduced the desired axioms for a globalness property of explainers. We then present a novel measure for this property of explainers and prove theoretically that this new measure satisfies all the desired axioms of explainer globalness.

---

### Decision · Program_Chairs · 2023-01-20

**Decision:**

Reject

**Justification For Why Not Higher Score:**

Writing flaws, weak experimental section.

**Justification For Why Not Lower Score:**

N/A

**Metareview: Summary, Strengths And Weaknesses:**

The paper proposes to use the Wasserstein distance to assess the locality/globality of an explainer. The paper received relatively poor grades overall.

A common thread in all reviews is that the paper is not primed for publication. That thread can either be found on criticism of the positioning of the paper w.r..t existing literature. I understand that the authors provide a few clarifying points below, but it is indispensable that they are integrated and re-reviewed. Another thread insists on the fact that the paper needs a rewrite, and that the experimental section is a bit weak.

I do believe that the current structure does not help indeed. I do find all sections up to (and including section 4) fairly limited in their contribution, since they are mostly reminders. They all conclude on the fact that it is interesting to compute the Wasserstein distance from uniform to explainers, but mostly recalling known facts (continuity, statistical complexity etc...) and take the time to mention alternative directions that are not explored. In that sense, that part reads like a mini-course, rather than an ICLR paper which should be straight to the point. Additionally, some parts are left out. For instance, the authors mention they use an entropic regularized W version. Is that the original W approx? the debiased approach? How do they tune $\varepsilon$ entropic regularization? I think this (and other) details would have mattered.

For these reasons I believe the draft is not ready, but i hope the feedback from reviewers can help them improve their submission.